# The Effect of Ionic Strength on the Formation and Stability of Ovalbumin–Xanthan Gum Complex Emulsions

**DOI:** 10.3390/foods13020218

**Published:** 2024-01-10

**Authors:** Yuanxue Gao, Wen He, Yan Zhao, Yao Yao, Shuping Chen, Lilan Xu, Na Wu, Yonggang Tu

**Affiliations:** 1Jiangxi Key Laboratory of Natural Products and Functional Food, Jiangxi Agricultural University, Nanchang 330045, China; ggy18985664004@163.com (Y.G.); hw19950506@163.com (W.H.); zhaoyan@jxau.edu.cn (Y.Z.); sunshineyy@jxau.edu.cn (Y.Y.); ncuchenshuping@sina.com (S.C.); xll15797927983@163.com (L.X.); tygzy1212@jxau.edu.cn (Y.T.); 2Agricultural Products Processing and Quality Control Engineering Laboratory of Jiangxi, Jiangxi Agricultural University, Nanchang 330045, China; 3Jiangxi Experimental Teaching Demonstration Center of Agricultural Products Storage and Processing Engineering, Jiangxi Agricultural University, Nanchang 330045, China; 4Nanchang Key Laboratory of Egg Safety Production and Processing Engineering, Jiangxi Agricultural University, Nanchang 330045, China

**Keywords:** ionic strength, stability of emulsions, ovalbumin, Xanthan gum

## Abstract

Protein–polysaccharide complexes have been widely used to stabilize emulsions, but the effect of NaCl on ovalbumin–xanthan gum (OVA-XG) complex emulsions is unclear. Therefore, OVA-XG complex emulsions with different XG concentrations at pH 5.5 were prepared, and the effects of NaCl on them were explored. The results indicated that the NaCl significantly affected the interaction force between OVA-XG complexes. The NaCl improved the adsorption of proteins at the oil–water interface and significantly enhanced emulsion stability, and the droplet size and zeta potential of the emulsion gradually decreased with increasing NaCl concentrations (0–0.08 M). In particular, 0.08 M NaCl was added to the OVA-0.2% XG emulsion, which had a minimum droplet size of 18.3 μm. Additionally, XG as a stabilizer could improve the stability of the emulsions, and the OVA-0.3% XG emulsion also exhibited good stability, even without NaCl. This study further revealed the effects of NaCl on emulsions, which has positive implications for the application of egg white proteins in food processing.

## 1. Introduction

Emulsions are usually thermo-dynamically and unstable polydispersed, formed by two incompatible liquids mixed under the action of external forces, in which the dispersed phase was dispersed in the continuous phase in the form of tiny droplets, which can be divided into oil in water (O/W), water in oil (W/O), and multiple emulsion structure systems. It has been widely used in various industries such as food, pharmaceutical, cosmetics, and agriculture. Currently, emulsions are often used in food products in the form of dairy products, sauces, meat products, etc. Many food additives or functional substances were also made into different types of emulsions for application in food [1]. However, since emulsions are thermo-dynamically unstable systems, they are susceptible to external environmental influences (e.g., pH, ionic strength, and temperature) during their processing, transportation, storage, and other applications, resulting in the separation of the aqueous and oil phases [2]. Although the design of appropriate two-phase interface parameters and the selection of suitable components stabilizers and other components prevented the separation of the aqueous and oil phases (such as surfactants, thickeners, gelling agents, and solid particles), they could only be stabilized by a small number of food-grade emulsifiers and thus severely restricted the development of emulsions in the food field. Therefore, it is necessary to study the stabilization mechanism of food-grade emulsions to improve their tolerance to environmental conditions during food processing.

In recent years, proteins have been widely used as emulsifiers because of their excellent biocompatibility, unique charge properties, large molecular structure, and amphiphilicity. They can be adsorbed to the surface of droplets during the homogenization process and stabilizing emulsions by generating electrostatic repulsion and strong spatial site resistance between particles [3]. Egg white proteins (EWP) mainly consist of ovalbumin (54%), ovotransferrin (12%), ovomucoid (11%), ovomucin (3.5%), and lysozyme (3.5%). These proteins are rich in non-polar amino acids and disulfide bonds, which can give egg whites good functional properties, such as emulsifying, gelling, and foaming properties, and have been widely used in the food industry [3]. Among them, ovalbumin (OVA), with a molecular weight of 45 kDa, is the most abundant protein in egg white protein, which consists of 385 amino acid residues, and its isoelectric point is between 4.5 and 4.8. Owing to the appropriate proportion of hydrophobic groups and hydrophilic groups, it can be quickly adsorbed to the oil droplet surface, thus forming a good emulsion [3,4]. However, the poor emulsifying property of OVA was mainly due to its poor thermal stability and emulsifying property near the isoelectric point, which limited the application of OVA emulsifiers in commercial products. To overcome these drawbacks, some rational molecular designs have been proposed to improve the emulsifying properties of OVA. For example, the addition of polysaccharides to form a complex with OVA improves their emulsification performance, as the complexes can form a relatively thick and highly charged surface layer, thereby reducing droplet interactions [4]. Although polysaccharides cannot be used as emulsifiers, ionic functional groups present in their molecules provide gelling and thickening properties. Thus, polysaccharides can adsorb to droplet surfaces via electrostatic interactions, thereby inhibiting the flocculation and aggregation of emulsion droplets [5]. Common polysaccharides include pectin, Xanthan gum, Arabic gum, etc. For instance, the complexes of EWP-high methoxylated pectin, which are synthesized via Maillard reactions, exhibit favorable emulsification properties, and demonstrate stability for a duration of 40 weeks at ambient temperatures [6]. This can be attributed to the heightened viscosity of the aqueous phase induced by pectin, as well as the exposure of hydrophobic protein groups via thermal treatment. Consequently, spatial repulsion and electrostatic interactions are facilitated, thereby enhancing the stability of the complex structure [6]. Xanthan gum is an anionic extracellular polysaccharide produced via the fermentation of *Xanthomonas campestris*. Its molecules are composed of D-glucose, acetyl, D-mannose, D-glucuronic acid, and pyruvic acid, forming a structural polymer with a repeating unit of five sugars. The molecular mole ratio is 28:17:3:2:(0.51–0.63), and the relative molecular mass ranges from 2 × 10^6^ to 5 × 10^7^ [7]. The effect of salt on food applications of Xanthan gum is primarily associated with the concentration of gum, ionic species, concentration, and ionic valence in foods. For instance, monovalent salts (NaCl) and divalent metal salts (MgCl_2_, CaCl_2_) cause a slight decrease in viscosity [7]. Meanwhile, hydrated Xanthan gum exhibits good salt resistance, allowing for the addition of up to 20–30% of salts without adversely affecting viscosity. Meanwhile, hydrated Xanthan gum exhibits good salt resistance, allowing for the addition of up to 20–30% of salts without adversely affecting the viscosity. Moreover, Xanthan gum is commonly used in the food, medicine, and chemical industries due to its stability, emulsification, thickening, suspension, and safety [7]. It was found that wheat gluten protein-XG complex-stabilized emulsions were found to have wider pH and salt content (0–1000 mM) stability than wheat gluten protein-stabilized emulsions alone, effectively protecting the encapsulated β-carotene from degradation [5]. The cereal protein-XG emulsions prepared under the same conditions remain stable at high salt concentrations, while cereal protein–pectin emulsions are stable only at low salt concentrations [8]. These studies show that XG had the potential to be used as a stabilizer in emulsions.

Protein–polysaccharide complexes have been widely used as media for the formation and stabilization of emulsions. However, protein–polysaccharide interactions are extremely complex, and environmental factors within the emulsion system can also influence the interactions, such as pH, ionic strength, polymer ratio, etc. [3]. Among them, pH and ionic strength were the most important factors. For example, pH affects the degree of dissociation of the charged groups in the complex, thus changing their positive or negative charges and numbers, which in turn affects the stability of the emulsion [9]. Chang et al. [10] found that the modulus of elasticity of the complex between rapeseed isolate protein and chitosan was at a pH of 6.0 and a mixing ratio of 16:1 to reach a maximum value, which, due to high electrostatic interactions and thick-walled sponginess, had a less porous microstructure. Meanwhile, ionic strength played an important role in the degree of electrostatic binding and structure of protein–polysaccharide complexes because ionic strength can induce and accelerate the phase separation of protein–polysaccharide systems [11]. It is mainly due to the fact that salt ions can shade the charges of protein molecules, reduce the intermolecular repulsive forces, and promote the cross-linking of proteins to form aggregates of larger particles. Thus, it can induce and accelerate the phase separation of protein–polysaccharide systems [11]. For example, the addition of salt ions to egg white protein/carrageenan and lysozyme/carrageenan mixtures inhibits the electrostatic interactions between the protein and the carrageenan, which in turn shifts the critical pH transition point to a lower pH value [3]. Furthermore, salt ions can compete with proteins or polysaccharides with charged groups, affecting protein–polysaccharide interaction forces. Niu Fuge et al. found that when NaCl was added to the mixture, the anion competed with the negative charge binding site of the Arabic gum chain to bind to OVA [12]. Likewise, the cation competes with the positive charge binding site of OVA for binding to Arabic gum, thus reducing the interaction between OVA and Arabic gum [12]. That is, in the presence of salt ions, there is a reduction in complex formation. Different concentrations of ionic strength have an effect on the electrostatic interactions between the egg white–polysaccharide, thus interfering with complex formation [5]. Therefore, it was necessary to investigate the effect of ionic strength on emulsion stability.

We established the egg white protein–polysaccharide gel and emulsion system in the early stage, compared the effects of neutral polysaccharides (dextran and inulin) and anionic polysaccharides (Xanthan gum) on the properties of egg white protein, and finally screened out Xanthan gum, which proved that EWP-XG gels [13] and OVA-XG complex emulsions [9] can be used as protein-based fat replacers in foods such as low-fat ice cream and mayonnaise. However, the properties of emulsions are influenced by various factors in food, such as pH, ionic strength, and temperature, which can lead to instability in their application, thereby limiting their practical applications [3,11,12]. Therefore, it is essential to study the impact of external factors. Because egg white proteins are a complex system, ovalbumin, the most abundant protein in egg whites, was chosen for mechanistic studies. Previously, we demonstrated the impact of pH and Xanthan gum concentration on the stability of the emulsions. Our findings revealed that the droplet size of the emulsions decreased gradually with increasing Xanthan gum concentration, and the storage stability of the emulsions significantly improved. Meanwhile, the stability of the emulsion largely depends on the pH value, and the OVA-XG composite emulsion exhibits better stability at pH 5.5 [9]. However, salt plays an important role in food processing, such as flavoring, preserving, and moisturizing. Meanwhile, the effect of NaCl on the stabilization of emulsions by OVA-XG complexes has not been determined. Therefore, the present study focused on investigating the impact of ionic strength on the formation and stabilization of OVA-XG complex emulsions. The study also aimed to explore the potential mechanism by which ionic strength affects the emulsion stabilization of this complex. The ultimate goal was to establish a theoretical foundation for creating fat substitutes for stabilized OVA-XG emulsions, developing nutrient delivery carriers, and advancing the overall utilization of poultry eggs.

## 2. Materials and Methods

### 2.1. Materials

Ovalbumin (OVA) from chicken egg white powder (62~88% pure by agarose electrophoresis, Cat. No. A5253) and XG (from *Xanthomonas campestris*, Cat. No. G1253) were purchased from Sigma-Aldrich (Shanghai, China). Soybean oil was obtained from Yihaijiali Investment Co., Ltd. (Shanghai, China). Nile red and Nile blue were provided by Ryon Biological Technology (Shanghai, China). For the preparation of all colloidal dispersions, deionized water was applied.

### 2.2. Emulsion Preparation

Referring to our previous studies [9], the stock solution of OVA and XG was mixed with distilled water to obtain an OVA solution concentration of 5 wt% and XG solution concentration of 0.5 wt%, and magnetically stirred for 2 h at room temperature, and then stored at 4 °C overnight for complete hydration. Different proportions of Xanthan gum were added to ovalbumin to form an OVA-XG complex solution with a certain concentration, in which the final concentration of OVA was wt%, and the proportions of XG were 0%, 0.1%, 0.2%, and 0.3%, respectively. Then, the complex solutions were magnetically stirred for 2 h at room temperature. The pH value of the complex solutions was adjusted to 5.5, and then the final OVA-XG complex solutions under different conditions were obtained after mixing with 0 M NaCl, 0.02 M NaCl, 0.04 M NaCl, and 0.08 M NaCl, which were used for the analysis of solution properties. Then, the OVA-XG emulsions were prepared by adding soybean oil (20%) into the final complex solutions then mixed for 10 s with a vortex machine and homogenized at 10,000 r/min for 8 min using a high-speed blender (UltraTurrax homogenizer, IKA T25 digital, Staufen, Germany) to obtain different types of emulsion.

### 2.3. Turbidity Measurements

The turbidity of the protein aggregate dispersions was measured at 600 nm using a UV-visible spectrophotometer [9]. The different OVA-XG complex solutions were diluted 10 times and then placed in a sample pool with a 1 cm optical path for further measurements. 

### 2.4. Ultraviolet (UV) Spectra Measurements

Ultraviolet spectra (UV) of different OVA-XG complex solutions were measured by a UV-5200PC spectrometer (MAPADA, Shanghai, China) [14]. Samples were diluted to a protein concentration of 0.1 wt% before measurement, and the scanning wavelength was set at 220–400 nm with a resolution of 0.5 RNN.

### 2.5. Fluorescence Spectra Measurements

The endogenous fluorescence of different OVA-XG complex solutions was determined by a 970 CRT Fluorescence spectrofluorometer (Shanghai, China) [9]. The samples were diluted 10 times with distilled water. The excitation wavelength was set at 280 nm, and the emission wavelength was set at 300–500 nm. The excitation and emission wavelength slits were both set at 5 nm.

### 2.6. Droplet Size Analysis

The average droplet size of OVA-XG emulsions was measured using a laser diffraction droplet size analyzer (Mastersizer 3000, Malvern Instruments Ltd., Worcestershire, UK). The samples were diluted to a concentration of approximately 0.001% *w*/*w* using distilled water prior to analysis, and the optical properties of the sample were defined as follows: the refractive indices of oil and water were 1.46 and 1.33, respectively, and the absorptivity was assumed to be 0. The droplet size was recorded as the volume-weighted mean diameter (d43) [15].

### 2.7. Zeta-Potential Measurements

The Zeta potential of different OVA-XG emulsions was determined using a particle electrophoresis instrument (Zetasizer Nano ZS90, Malvern Instruments Ltd., Worcestershire, UK) [16]. In order to reduce the effect of multiple light scattering, samples should be diluted to a concentration of about 0.001% *w*/*w* before determination and then loaded into a cuvette with electrodes and equilibrated for 60 s at 25 °C.

### 2.8. Interfacial Tension Measurements

In order to reduce the interference of soybean oil, purification is necessary before use. The purification method was conducted as follows [17]: 8 g of Florisil molecular sieve is dispersed into 200 mL of soybean oil and stirred with a magnetic stirrer. The mixture is then centrifuged to remove the Florisil molecular sieve. A new molecular sieve is added, and the process is repeated 3–4 times until the surface tension of ultrapure water in the soybean oil does not change.

The dynamic interfacial tension between the purified soybean oil and the OVA-XG composite solution was measured at room temperature (25 °C) using an optical surface analyzer (OSA 100, Ningbo Lauda Technology Co., Ltd., Ningbo, China). Under video camera supervision, a suitable quantity of purified soybean oil was poured into a cuvette as the bulk oil phase. The tip of a cylindrical syringe containing the OVA-XG compound solution was then immersed, allowing a droplet of water to enter the oil phase from the syringe tip. Data points were recorded every second for 30 min over a constant area of 15 mm^2^ of the drop [17]. 

### 2.9. Rheological Property Measurements

The rheological analysis was performed by DHR-1 rheometer (TA Instruments, Newcastle, DE, USA) with a concentric cylinder (diameter = 28 mm) [18]. Briefly, 20 mL samples were placed in the concentric cylinders with the test temperature of 25 °C and waited for 3 min to balance the temperature of the emulsion in the concentric cylinders. The frequency sweep process was from 0.1 to 100 (rad/s) at a fixed strain of 10% that was within a linear viscoelastic range. The apparent viscosity was recorded as the shear rate was increased from 0.1 to 100 s^−1^. And the Peltier temperature control system was utilized to maintain the temperature at 25 °C for all samples.

### 2.10. Creaming Index

Fresh OVA-XG emulsions were placed in the centrifuge tube with 0.02% sodium azido to inhibit the growth of microorganisms. Then, the emulsion was stored at 25 °C for 7 days, and the layered emulsions were observed. The total height of the emulsions (He) and the height of the serum layer (Hs) were measured [15]. The emulsions stability was characterized by creaming index (CI), and the calculation formula is as follows: CI (%) = (HS/HE) * 100%

### 2.11. Confocal Laser Scanning Microscopy (CLSM)

The microstructure of different OVA-XG emulsions was measured using a confocal laser scanning microscope (Carl Zeiss LSM710, Jena, Germany). Nile blue dissolved in water and Nile red dissolved in ethanol were prepared into 0.1% (*w*/*v*) Nile red dye and 0.1% (*w*/*v*) Nile blue dye, and then wrapped in aluminum foil and stored away from light. Briefly, a 2 mL emulsion was mixed with 50 μL dye, mixed with a vortex machine, and reacted for 30 min in the dark. An appropriate amount of the stained emulsion was taken on a glass slide and observed using CLSM. Argon and He/Ne lasers are used as excitation sources, and the excitation wavelength is set to 488 nm and 633 nm, respectively. Microscopic images of OVA-XG emulsions were collected to observe the distribution of oil droplets (red) and proteins (green) [16].

### 2.12. Statistical Analysis

All measurements were conducted at least three times in parallel, and the average value was taken. The results were expressed as the calculated means and standard deviations. The statistical analysis was carried out by SPSS 20.0 (SPSS Inc., Chicago, IL, USA) with a significant difference between means (*p* < 0.05) using Duncan’s multiple analysis. The graphs were created with Origin (Origin Lab, Northampton, MA, USA).

## 3. Results and Discussion

### 3.1. Turbidity of Different OVA-XG Composite Solutions

Turbidity can reflect the degree of protein aggregation as well as the droplet size of proteins, so it is widely used to characterize the sensitivity of protein-based nanoparticles to salt concentration [19]. In order to investigate the effect of ionic strength on the interaction of OVA-XG composite solutions, the turbidity was analyzed. Figure 1 illustrates the effect of different ionic strengths (0, 0.02 M, 0.04 M, and 0.08 M NaCl) on the turbidity of different OVA-XG composite solutions (XG concentrations of 0, 0.1, 0.2, and 0.3 wt%) under the condition of pH 5.5. When NaCl was not added, the turbidity of the composite solution decreased first, increased, and then decreased with the increase in XG concentration. However, in the presence of NaCl, the turbidity showed a completely opposite trend with the increase in XG concentration. At low concentrations of XG, turbidity increased with the increase in NaCl concentration, while at high concentrations of XG, turbidity decreased. It may be that when pH was 5.5, OVA molecules were negatively charged, and XG was also a negatively charged polysaccharide. The complexes of OVA and XG were mainly dominated by electrostatic repulsion.

When XG concentrations were less than 0.2 wt%, the increase in salt ions significantly shielded the charges of OVA and XG molecules, making the electrostatic repulsion between them weaker and thus promoting the formation of complexes. At high XG concentration, the turbidity value decreased with the increase in salt ion concentration, i.e., the complex formation was weakened, which was mainly attributed to the fact that the high concentration of XG increased the viscosity of the system and increased the spatial site resistance of the system, thus inhibiting the formation of complexes [7].

### 3.2. UV and Fluorescence Spectra of OVA-XG Composite Solutions

The study investigated the interaction between proteins and polysaccharides using UV and fluorescence spectroscopy to confirm the structural changes in proteins under different ionic strengths. The absorption spectral intensities and peak wavelength changes were used to determine the intensity and mechanism of their interaction.

The UV absorption peaks of proteins were mainly attributed to aromatic amino acids, namely tryptophan, tyrosine, and phenylalanine. The extent of displacement of the amino acid UV absorption peaks can be used to infer the conformational changes in proteins [20]. In order to clarify the influence of ionic strength on the interaction of the OVA-XG solution system, the protein conformation and UV absorption spectra were analyzed in this paper. However, since the UV absorption peaks of these three amino acids at 280 nm may overlap with each other and be difficult to distinguish, a second-order derivative analysis of the UV absorption peaks was conducted. As shown in Figure 2, the UV absorption peaks of the OVA-XG composite solution shifted to varying degrees at different ionic strengths as the concentration of XG increased. This suggests that the interaction between OVA and XG influenced the protein molecule’s microenvironment, causing the displacement of amino acid residues. In the absence of XG, the maximum absorption peaks of UV spectra of OVA-XG emulsions were red-shifted with increasing NaCl concentration, indicating a conformational change in OVA and the movement of amino acid residues to a more hydrophobic environment; this phenomenon has also been observed in potato protein–chitosan complexes [14]. However, as the XG and NaCl concentrations increased, the maximum absorption peak was blue-shifted, which may be attributed to the interaction between OVA and XG changed the microenvironment of protein molecules, resulting in more exposure of tyrosine residues in a hydrophilic environment [12]. From Figure 2, it can be observed that the addition of a small amount of salt ions (0.02 M) leads to an increase in the absorbance of the solution. It is possible that the protein molecule unfolds, leading to an increase in the chromogenic groups on the surface and, subsequently, an increase in absorbance.

The excitation of aromatic amino acids in protein molecules changes the fluorescence intensity and maximum absorption wavelength. When the maximum absorption wavelength (λ max) of the fluorescence emission peak shifts to a longer wavelength, it indicates that the amino acid residues are red-shifted, meaning that more amino acid residues are exposed to the hydrophilic environment. Conversely, a blue shift occurs when tryptophan moves to a more hydrophobic environment [21]. Figure 3 shows the fluorescence spectra of the OVA-XG complex solutions system at different ionic strengths. From Figure 3, it can be seen that the maximum absorption wavelengths of the fluorescence emission peaks were all located between 325 and 350 nm. From Figure 3a, it can be seen that without XG, with the increase in NaCl concentration, the maximum fluorescence emission peak was red-shifted, and the fluorescence intensity increased from 853 nm to 911 nm, indicating that at pH 5.5, OVA was negatively charged and intermolecular repulsion was strengthened, the addition of NaCl caused the intermolecular structure of the protein to unfold, and tryptophan was exposed to the protein surface and moved to a more hydrophilic environment, leading to an increase in the polarity of its environment and a continuous increased in the maximum absorption peak [22]. As shown in Figure 3b–d, in the presence of XG, the maximum fluorescence absorption peaks changed with the addition of different concentrations of NaCl, and the fluorescence emission maxima were all reduced, and the λ max was blue-shifted. The λ max of the 0.1% XG-OVA, 0.2% XG-OVA, and 0.3% XG-OVA complex solution systems changed from 343 nm to 331 nm, from 341 nm to 336 nm, and from 339 nm to 337 nm, respectively. This wavelength range was a typical distribution of tryptophan residues in a relatively hydrophobic environment; that is, the addition of XG and NaCl changed the protein conformation to varying degrees, the microenvironment of amino acids changed, and tryptophan moved to a more hydrophobic environment [23]. This result corresponds to the second-order UV conduction result. Because the addition of polysaccharides increased intermolecular interactions due to the collision between protein–polysaccharide molecules, some of the tryptophan residues were wrapped within other macromolecular side chains. Secondly, after adding polysaccharides and salt, the protein structure was disrupted to some extent. This caused the hydrophobic amino acids to be exposed and gathered on the protein surface, leading to the exposure of the chromogenic group to the solvent and a subsequent reduction in fluorescence intensity [24].

### 3.3. Droplet Size of OVA-XG Complex Emulsions

The variation in droplet size is an important method to evaluate the stability of emulsions [25], and usually, the smaller the droplet size of an emulsion, the better its stability. As shown in Figure 4a, the addition of salt ions will alter the droplet size of the emulsion. Adding NaCl (0.02–0.08 M) to OVA, OVA-0.1% XG, and OVA-0.2% XG emulsions can significantly reduce the droplet size of the emulsion, which were more homogeneous and stable compared to those without NaCl. The effect of salt concentration varied; the addition of 0.04 M salt caused the lowest droplet size reduction in OVA and OVA-0.1% XG emulsions, decreasing from 33.3 μm to 26.6 μm and from 27.4 μm to 21.8 μm, respectively. While in OVA-0.2% XG emulsions, the smallest droplet size of 18.3 μm was caused by the 0.08 M NaCl. It may be that different salt concentrations had different effects on the electrostatic interaction, ion bridging, or ion binding in OVA-XG emulsions prepared with different XG concentrations. However, within a specific range of ionic strength, these interactions cause the emulsion to flocculate rapidly and form a network structure to stabilize the emulsion [26]. At the same time, moderate flocculation is beneficial for the stability of the emulsion system [27]. Additionally, the addition of salt can also enhance the structural properties of the protein. It can enhance the rigidity and thickness of the interface layer, thereby improving the interfacial stability of the emulsion [26]. Interestingly, however, the droplet size of the OVA-0.3% XG emulsion gradually increased with the addition of salt ions (0.02–0.08 M), from 18.4 μm to 24.4 μm. One possible reason is that the salt neutralizes a large amount of negative charge on the droplet surface, reducing the surface charge density and forming an electrostatic shielding effect around the droplet. The electrostatic repulsion between droplets was reduced, and their hydrophobic interaction was enhanced, which in turn led to the aggregation of droplets, resulting in an increase in the average droplet size of the emulsion with salt ion concentration [28]. Moreover, the addition of salt ions alters the osmotic pressure change in the system, which in turn affects the droplet size [29]. For example, osmotic pressure can reduce the swelling capacity of OVA-XG complexes in salt solutions, resulting in increased aggregation and precipitation [29].

From Figure 4a, the average droplet size of the emulsions without NaCl and with NaCl (0.02–0.04 M) decreased gradually with the increase in XG concentration, probably because the continuous addition of XG increased the viscosity of the emulsion system and the migration rate and aggregation of droplets were slowed, thereby improving the emulsion stability [9,30]. Meanwhile, when the pH value is 5.5, OVA becomes negatively charged, and the combination with negatively charged XG increases the electrostatic repulsion between droplets and weakens the electrostatic interaction, which, in turn, causes the droplet size to decrease. Similar results were reported that the droplet size of zein-XG (0.2–1%) complexes and tea protein-XG (0–2.5%) complexes [31] both decreased with increasing XG concentration. However, the droplet size of the emulsions with the added salt concentration of 0.08 M showed a trend of decreasing first and then increasing with the concentration of XG. At high salt concentrations, it may be that the excess XG increases the amount of unabsorbed content in the continuous phase, which in turn triggers bridging flocculation between emulsion droplets, resulting in protein emulsion destabilization [32].

### 3.4. Zeta-Potential of OVA-XG Complex Emulsions

The electrostatic interaction between charged biopolymers in an aqueous solution was the primary driving force for the formation of complex coacervates [33]. Therefore, emulsion stability can be achieved by adjusting the charge distribution on the surface of emulsion droplets [16]. Zeta potential is typically used to indicate the strength of the mutual attraction or repulsion between emulsion droplets [34]. As shown in Figure 4b, the addition of NaCl (0–0.08 mol/L) decreased the zeta potential of all samples and was negative, indicating that the droplet surface was negatively charged. It was found that as the concentration of NaCl increased, the zeta potential value of the emulsion stabilized by OVA alone decreased from −24 mV to −29 mV, and its absolute value increased. The emulsion became more stable, indicating that the addition of NaCl provided some electrostatic stabilization to the emulsion droplets. It has been reported in the literature that emulsions exhibit good stability when the absolute zeta potential values exceed 30 mV. This is attributed to increased electrostatic repulsion between emulsion droplets, effectively preventing coalescence formation [35]. The zeta potential values of OVA-XG complex emulsions were found to decrease with increasing NaCl concentration under the same XG addition conditions (*p* < 0.05). This may be attributed to the negative charge of XG and OVA at pH 5.5, which increased the electrostatic repulsion between droplets and weakened the electrostatic mutual attraction effect. Meanwhile, the electrostatic shielding effect of NaCl would further strengthen this force and prevent droplet aggregation [30].

Polysaccharides can interact with proteins via electrostatic or hydrophobic forces, thereby influencing the charge distribution of emulsions and subsequently impacting their stability. As shown in Figure 4b, the absolute potential value of all emulsions increased with XG concentration under salt ion concentrations ranging from 0 to 0.08 mol/L. This could be attributed to the gradual increase in XG led to the distribution of a large number of polysaccharides with negatively charged groups in the emulsion. This increased the electrostatic repulsion of the molecules and effectively prevented the proximity and aggregation of the emulsion particles, thereby enhancing the stability of the emulsion [36]. And the incorporation of the macromolecular polysaccharide XG enhanced the viscosity of the OVA emulsion. Within this system, XG and OVA complexes formed a weak gel network structure that restricts the flow and coalescence of lipid droplets [26]. Secondly, XG is an anionic polysaccharide with a potential value of approximately −45 mV in a saline solution. Therefore, excessive or high concentrations of XG may result in a reduction in the system’s potential. However, it was observed that the OVA-0.3% XG emulsion exhibited the lowest potential (all lower than −55 mV) at each ion concentration, possibly because of the presence of free XG alongside XG bound to protein in the system. Yet the potential represented the average potential of free OVA, XG, and OVA-XG complexes. In addition, the literature has reported an increase in the absolute value of protein–polysaccharide complex potential at high salt concentrations. However, their stability was poor, as similar results had been observed in chitosan and casein complexes [31]. This result aligns with the previously mentioned pattern of variation in droplet size.

### 3.5. Interfacial Tension of OVA-XG Complex Solutions

It is known that proteins reduce the interfacial tension of the oil–water interface because of the amphiphilic nature given by their structure, and the effective degree of protein adsorption at their interface and the stability of the emulsion can be inferred from the change in interfacial tension. Generally, a smaller interfacial tension indicates a more stable emulsion [37]. As shown in Figure 5, the dynamic interfacial tension of all samples showed a trend of rapid decrease and then equilibrium with increasing time. This suggests that the protein emulsions reduced the interfacial tension in a consistent manner, even at varying NaCl concentrations. When no XG was added, the increase in NaCl concentration had little effect on the change in interfacial tension of the OVA solution. However, when 0.1% and 0.2% XG were added, the interfacial tension of the OVA-XG solutions all showed a decreasing trend as NaCl concentration increased. The composite emulsions reached their lowest interfacial tension value at a NaCl concentration of 0.08 mol/L, which may be attributed to the disordered structure of the protein at the oil–water interface led to a rapid change in its conformation with increasing NaCl concentration, which reduced the interfacial tension and produced a lower equilibrium interfacial tension over time [38]. Secondly, due to the hydrophilicity of OVA, the addition of salt ions significantly reduced the dispersion of the emulsion extremely low, which led to the gradual stabilization of the composite solution and resulted in lower energy of the entire system [39]. Furthermore, the inclusion of NaCl, comprising sodium and chloride ions, reduced the electrostatic repulsion between charged emulsions at the oil–water interface. This improved the stability of the protein adsorption layer and allowed NaCl and low concentrations of XG (0.1 wt%, 0.2 wt% to work together to decrease the interfacial tension of the composite solutions synergistically [40]. However, the interfacial tension of the OVA-0.3% XG solutions increased with higher NaCl concentration, likely because of the elevated XG concentration and the formation of insoluble polymers by OVA-0.3% XG. The excessive XG hindered the adsorption of proteins at the oil–water interface, leading to an increase in interfacial tension [32].

Under the influence of salt-free ions and a low salt concentration of 0.02 M, the interfacial tension of OVA-XG composite solutions showed a trend of increasing and then decreasing with the concentration of XG, indicating that the addition of XG could promote protein adsorption to some extent under the salt-free and low salt conditions. However, under the influence of salt concentrations of 0.04 M and 0.08 M, the interfacial tension of OVA-XG complex solutions gradually increased with an increase in XG concentration, which was attributed to the fact that the free XG at this ionic strength increased the viscosity of the continuous phase and inhibited the adsorption behavior of the complex at the oil–water interface, so the surface tension increased [41].

### 3.6. Rheological Properties of OVA-XG Composite Emulsions

Rheology is an important physical and chemical indicator for studying the texture, taste, and structure of food products. Figure 6 shows the stable flow characteristics of the emulsions prepared using OVA-XG with varying concentrations of NaCl at pH 5.5. In the range of 0.1~100 s^−1^ shear rate, the apparent viscosity of all samples decreased as the shear rate increased. The composite emulsions exhibited the shear thinning behavior characteristic of pseudoplastic fluids [42]. Shear-thinning behavior is a common characteristic of emulsions, and this property may be due to the application of shear stress, which changes the size and shape of the oil droplets in the flocculated state under the shear action [43]. At a lower shear rate, the viscosity of the OVA emulsion increased after the addition of NaCl, indicating that the interaction between the emulsion particles was enhanced due to the addition of sodium ions (Na^+^) [22]. The Na^+^ shielded the negative charge on the protein surface, reducing the intermolecular electrostatic repulsion and promoting hydrophobic interactions, which led to the protein particle aggregation [22]. Also, the flocculation phenomenon between emulsion droplets led to an increase in emulsion viscosity. Interestingly, at low XG concentrations, the viscosity of OVA-0.1% XG emulsions formed with small amounts of salt ions (0.02 M, 0.04 M) showed a decreasing trend, while the viscosity of emulsions at high ion concentrations (0.08 M) increased. This indicates that the composite emulsion formed by OVA-0.1% XG had a higher degree of inter-droplet flocculation at high salt concentrations compared to a low salt concentration. As the interaction force between droplets was reduced, small amounts of salt ions caused the emulsion viscosity to decrease. However, the increase in ion concentration leaded to an increase in emulsion viscosity and the formation of a denser gel network structure [18].

The apparent viscosity of the composite emulsion increased significantly with the increase in XG concentration, especially when the XG concentration reached 0.3 wt%; the apparent viscosity of the emulsion increased by an order of magnitude, while the salt ion concentration did not have a significant effect on the viscosity of the composite emulsion with the increase in shear rate. As shown in Figure 6, with the increase in salt ion concentration, the shear viscosity in OVA-0.2% XG and OVA-0.3% XG emulsions exhibited roughly the same trend, showing the shear phenomenon of shear thinning. It was found that the ionic strength of the OVA-XG emulsion system had a minimal impact on the viscosity of the emulsion system, while XG greatly enhanced its viscosity. This was because as the concentration of XG in the system increased, the electrostatic repulsion between XG molecules increased, and the interaction between XG and OVA molecules, i.e., electrostatic gravitational force and hydrophobic interaction, also increased. The force between the droplets increases, resulting in an increasing viscosity of the emulsion [9]. This indicates that the apparent viscosity of the OVA-XG complex emulsion is dependent on the concentration of XG. A sufficiently high amount of XG can inhibit OVA settling and aggregation, which is crucial for the practical application of the OVA-XG complex system, such as in fat substitution.

### 3.7. The Creaming Index of OVA-XG Complex Emulsions

The creaming index (CI) can be used to evaluate the equilibrium state of the two phases of the emulsion during storage. The higher the CI value (%), the more the droplets coalesce, and the higher the proportion of the water layer formed in the emulsion, the more unstable the emulsion; however, it is generally believed that the optimal CI value for better emulsion stability is between 5 and 35% [44]. As shown in Figure 7, the CI values in emulsion storage were not linearly related to the concentration of NaCl. The addition of salt to OVA and OVA-0.1% XG emulsions was found to reduce their CI values, indicating that salt ions can enhance the stability of emulsions. But, its CI was still greater than 47%, indicating that the stability of the emulsion prepared under this condition was still poor. However, in the OVA-0.2% XG emulsion, the addition of 0.04 M and 0.08 M salt caused its CI to be lower than 9%, while the addition of low salt 0.02 M caused a significant increase in its CI to 14.4%, which may be due to the electrostatic shielding effect of this salt ion concentration that decreased the repulsive force between droplets and the aggregation of the emulsion [29]. It is noteworthy that when various salt concentrations were added to the OVA-0.3% XG emulsion, the CI values were consistently 0 for each salt ion concentration. This suggests that at a higher XG concentration (0.3%), a network gel structure with high viscosity is formed for the OVA-XG composite emulsion, binding the fat globules into the network, and hindering the movement of oil droplets. In contrast, salt ions have less effect on the CI value of this composite emulsion.

Compared with ionic strength, the concentration of XG has a greater influence on the CI value of the OVA-XG emulsion. As shown in Figure 7, compared with the addition of XG, the CI value of a single OVA emulsion was the largest after 7 days of storage at 25 °C, indicating the most severe delamination and the poorest stability; this was because the emulsion was a thermo-dynamically unstable system. During storage, the molecular movement within the emulsion system accelerates, leading to the rearrangement or detachment of the protein molecules adsorbed at the oil–water interface, thereby altering the interfacial structure. This leads to a broader droplet size distribution and increases the probability of coalescence [45]. However, at the same ionic strength, the CI value of the emulsion decreased significantly with the increase in XG concentration. When XG was added at 0.1 wt%, the CI value decreased more slowly; when the XG concentration was 0.2 wt%, the CI value decreased from 70% to 10%, indicating that the stability of the emulsion was good, probably due to the weak gel structure formed by the presence of XG [46], which hinders the movement of droplets. Interestingly, XG showed excellent stability without phase separation (i.e., 0% CI) at a concentration of 0.3 wt%, probably due to the abundance of a large amount of unbound XG, which increased the viscosity of the continuous phase and formed a protective layer around the droplets, reducing inter-droplet collisions and resulting in a CI of 0 [26].

### 3.8. Microstructures of OVA-XG Complex Emulsions

Microstructure analysis can visualize the distribution of fat globules and the droplet size of the emulsions. Figure 8 showed confocal laser scanning microscopy (CLSM) of the emulsion prepared by OVA-XG under different ionic strengths, in which the oil phase was marked in red, the protein marker in the aqueous phase was in green, and the green shiny area was the protein enriched area. As shown in Figure 8, the microstructure of the single OVA emulsion revealed large fat globules with unevenly distributed gaps between lipid droplets, preventing the formation of a dense network structure and resulting in poor emulsion stability. As the concentration of salt ions increases, the large oil droplets gradually become smaller. This is likely due to the synergistic effect of OVA and NaCl, which reduces the interfacial tension and forms an interfacial protein film around the oil droplets. Consequently, this reduces the collision between molecules at the interface [40,47]. When 0.1% and 0.2% XG were added, the droplet size of the OVA-XG composite emulsions decreased. Additionally, with the increase in NaCl addition, the emulsion generally showed a trend of decreasing oil droplets, which may be due to XG and OVA forming a denser three-dimensional network structure and a stable, protective layer at the oil–water interface or because the addition of NaCl led to an increase in the absolute value of zeta potential and the dissociation of OVA-XG aggregates [48]. However, when XG was added at 0.3%, the oil droplets gradually increased with the increase in NaCl. In addition, a significant flocculation of the emulsion droplets was observed. This phenomenon can be attributed to the high concentration of NaCl, which weakens the electrostatic repulsion between droplets by shielding a portion of the protein surface charge and promotes the collision between droplets, leading to the presence of incompletely wrapped oil droplets in the emulsion [27].

In general, the smaller the size of oil droplets in an emulsion, the greater its stability. Figure 8 shows the droplets of emulsions without and with the addition of salt ions of 0.02 M and 0.04 M decreased with the increase in XG concentration, indicating that XG can be used as a stabilizer for emulsions. It may also be that low and medium concentrations of salt further strengthen the network structure of OVA-0.1% XG, OVA-0.2% XG, and OVA-0.3% XG, wrapping many oil droplets within, limiting the migration and aggregation of droplets. However, when the salt ion concentration was 0.08 M, the droplet size of the emulsion and the concentration of XG change irregularly, and the fat particles in OVA-0XG and OVA-0.2% XG emulsions were small and widely distributed, but the oil droplets in OVA-0.1% XG and OVA-0.3% XG were larger and more dispersed, probably because the high concentration of salt ions broke the OVA-XG at different concentrations of XG interactions, which in turn led to different structures of the complex emulsions. This was consistent with the results of the findings from the droplet size analyzer (Figure 4a) and the potentials (Figure 4b).

## 4. Conclusions

In this paper, the effects of different ionic strengths and XG concentrations on OVA-stabilized emulsions were investigated. The results showed that the addition of different concentrations of salt ions increased or decreased their electrostatic or hydrophobic interactions and promoted or inhibited protein refolding and aggregation, thereby affecting the OVA-XG complex solution. Secondly, the addition of XG can increase the viscosity, salt can improve its interfacial adsorption capacity, and XG and salt can synergistically improve the stability of the emulsion under specific conditions. In the presence of XG, the rheology of OVA-XG composite emulsions was characterized by shear thinning properties and a small amount of salt ions weakens the interaction force between droplets. With the increase in salt concentration, the network structure formed inside the emulsion, the absolute value of zeta potential increased significantly, the droplet size of the emulsion stabilized by the OVA-XG complex gradually decreased, and the seven-day storage stability of the emulsion improved. In summary, our results confirm that ionic strength and XG concentration can significantly affect the stability of emulsions, with OVA-0.2% XG composite emulsions being the most stable at a salt concentration of 0.08 M and OVA-0.3% XG at no salt. Our study provides information for the processing of egg white protein-derived foods.

## Figures and Tables

**Figure 1 foods-13-00218-f001:**
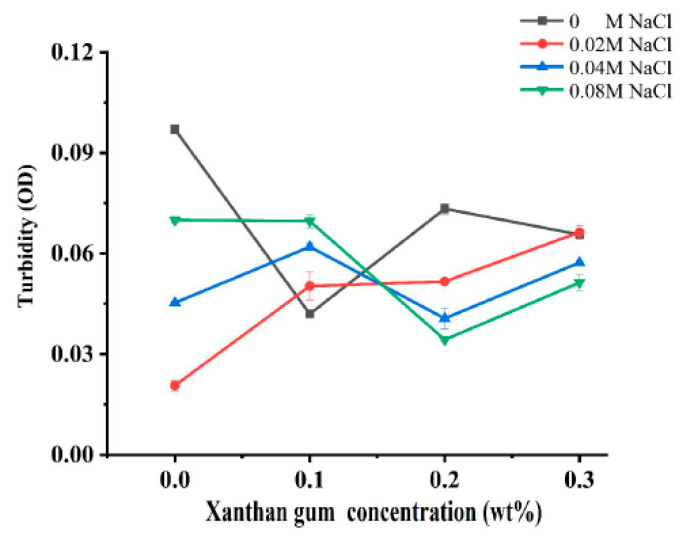
Effect of different ionic strength (0, 0.02 M, 0.04 M, 0.08 M NaCl) on the turbidity of different OVA-XG composite solutions (XG concentration of 0, 0.1, 0.2, 0.3 wt%) under the condition of pH 5.5.

**Figure 2 foods-13-00218-f002:**
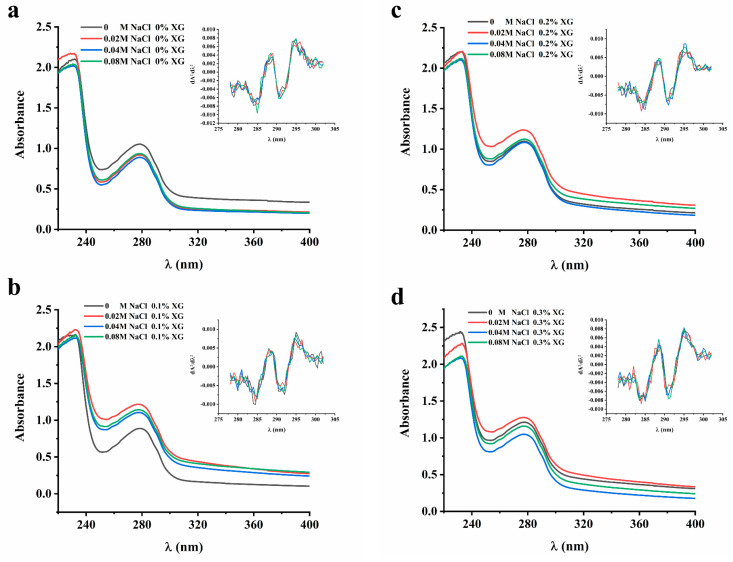
Effect of different ionic strength (0, 0.02 M, 0.04 M, 0.08 M NaCl) on the UA spectrum of different OVA-XG composite solutions. (**a**): different NaCl, 0 wt% XG; (**b**): different NaCl, 0.1 wt% XG; (**c**): different NaCl, 0.2 wt% XG; (**d**): different NaCl, 0.3 wt% XG.

**Figure 3 foods-13-00218-f003:**
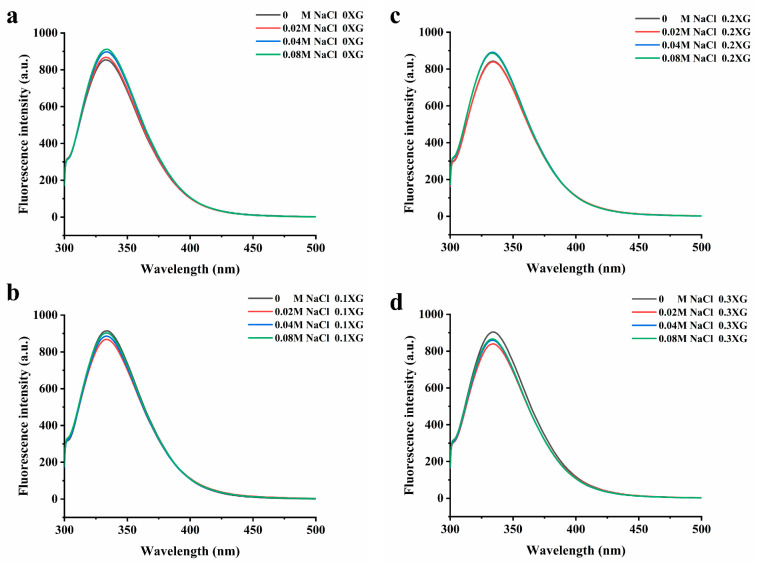
Effect of different ionic strength (0, 0.02 M, 0.04 M, 0.08 M NaCl) on the fluorescence spectrum of different OVA-XG composite solutions. (**a**): different NaCl, 0 wt% XG; (**b**): different NaCl, 0.1 wt% XG; (**c**): different NaCl, 0.2 wt% XG; (**d**): different NaCl, 0.3 wt% XG.

**Figure 4 foods-13-00218-f004:**
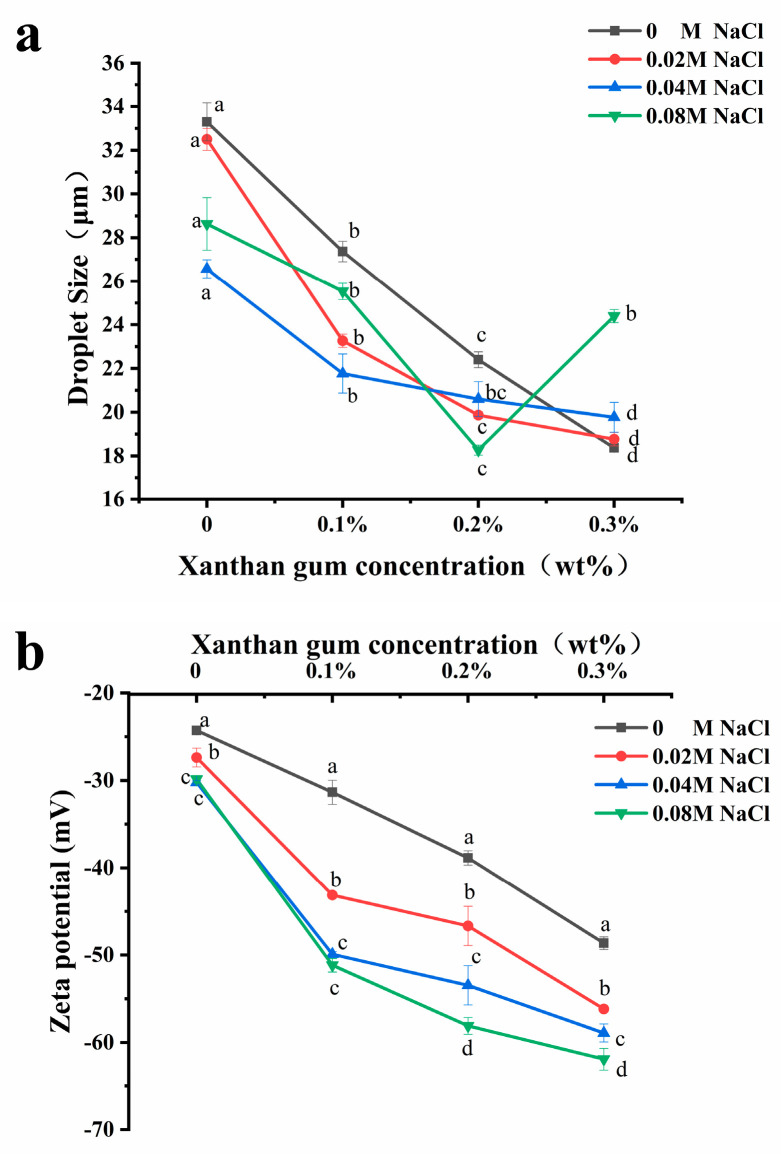
Effect of different ionic strength (0, 0.02 M, 0.04 M, 0.08 M NaCl) on the droplet sizes (**a**) and the zeta potential (**b**) of different OVA-XG complex emulsions (XG concentration of 0, 0.1, 0.2, and 0.3 wt%) under the condition of pH 5.5. Same letters indicate no significant difference, while different letters indicate a significant difference (*p* < 0.05).

**Figure 5 foods-13-00218-f005:**
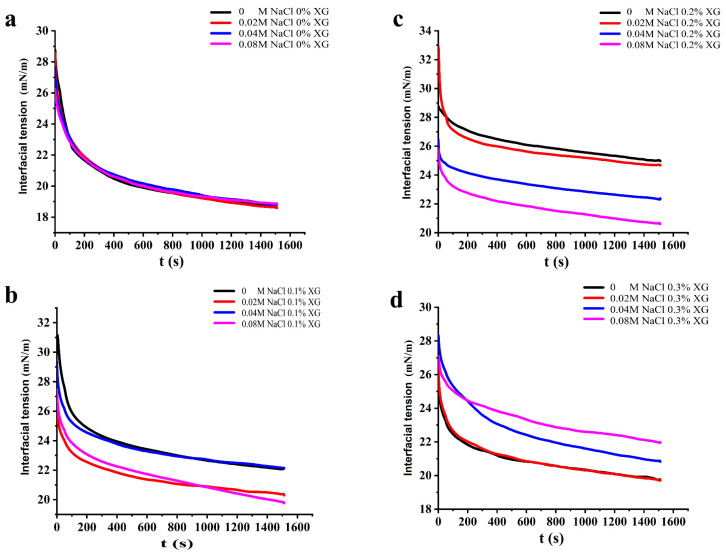
Effect of different ionic strengths (0, 0.02 M, 0.04 M, 0.08 M NaCl) on the interfacial tension of different OVA-XG complex emulsions. (**a**): different NaCl, 0 wt% XG; (**b**): different NaCl, 0.1 wt% XG; (**c**): different NaCl, 0.2 wt% XG; (**d**): different NaCl, 0.3 wt% XG.

**Figure 6 foods-13-00218-f006:**
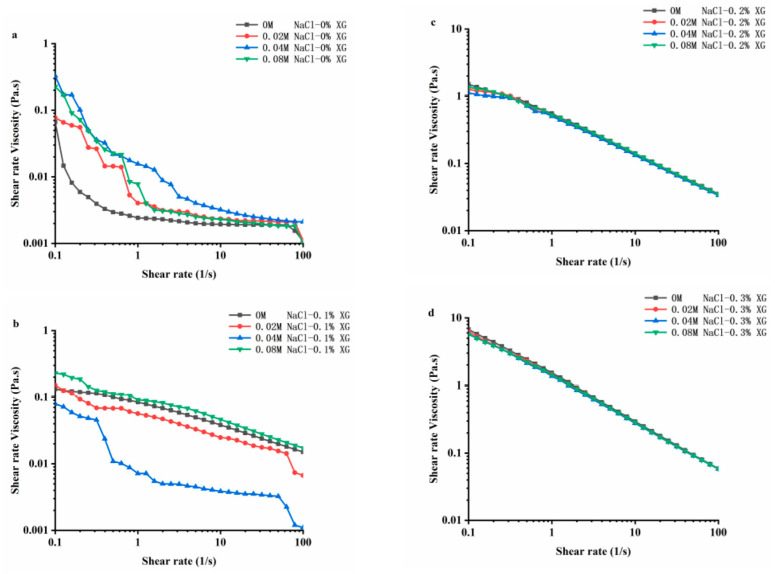
Effect of different ionic strengths (0, 0.02 M, 0.04 M, 0.08 M NaCl) on the rheology property of different OVA-XG complex emulsions. (**a**): different NaCl, 0 wt% XG; (**b**): different NaCl, 0.1 wt% XG; (**c**): different NaCl, 0.2 wt% XG; (**d**): different NaCl, 0.3 wt% XG.

**Figure 7 foods-13-00218-f007:**
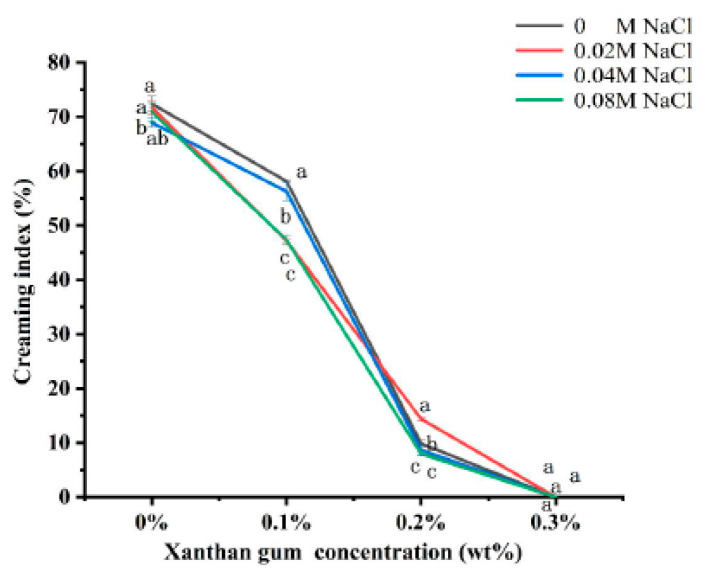
Effect of different ionic strength (0, 0.02 M, 0.04 M, 0.08 M NaCl) on the creaming index of different OVA-XG complex emulsions (XG concentration of 0, 0.1, 0.2, 0.3 wt%) under the condition of pH 5.5. Same letters indicate no significant difference, while different letters indicate a significant difference (*p* < 0.05).

**Figure 8 foods-13-00218-f008:**
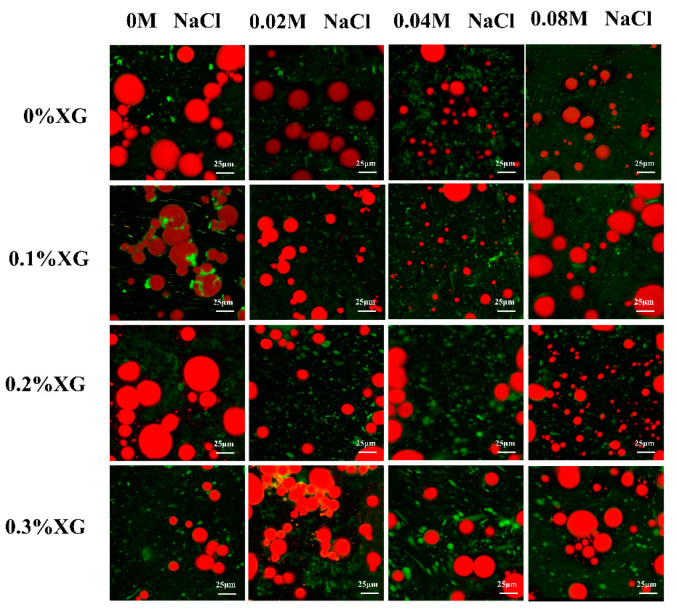
Effect of different ionic strengths (0, 0.02 M, 0.04 M, 0.08 M NaCl) on the confocal laser scanning microscopy of different OVA-XG complex emulsions (XG concentration of 0, 0.1, 0.2, 0.3 wt%) under the condition of pH 5.5.

## Data Availability

Data is contained within the article.

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
