# Peer review of "The Effect of Ionic Strength on the Formation and Stability of Ovalbumin–Xanthan Gum Complex Emulsions"

_foods, 2024, doi:10.3390/foods13020218_

Round 1

Reviewer 1 Report

Comments and Suggestions for Authors

The purpose of this study was to determine the effect of ionic strength on emulsion formation and stabilization by the ovalbumin-xanthan gum complex and explored the potential mechanism of ionic strength on emulsion stabilization by this complex, in order to providing a theoretical basis for the overall development of poultry eggs.

The research topic is relevant to colloid issues.

Some corrections:

Line 29; thermo-dynamically

According to the authors and cited references the theme can provide innovative information.

The methodology is well explained.

The conclusions are in accordance with the proposed objectives and with the findings of the project.

The figures presented in the text are well presented and provide information on important points of the project. Some corrections are necessary:

Figure 6 -- correct:

Shear rate Viscosity (Pa.s)

Author Response

Dear Reviewer,

Thank you very much for your critical and valuable comments of reviewers concerning our manuscript entitled “Effect of ionic strength on the formation and stability of the ovalbumin-xanthan gum complex emulsions” (ID: foods-2753995). We have carefully revised the manuscript. All changes to the text have been made in the revised manuscript marked with red, so that you  may easily identify them. We also responded point by point to you and reviewer’s comments as listed below. We hope the revised paper would satisfy you and the reviewer. We would appreciate you very much if you let us know your decision at your earliest convenience.

Thank you very much again for your kind help.

Best wishes,

Na Wu

Response to reviewers:

Point 1. Line 29; thermo-dynamically

Response: Thank you for your considerate review, and we apologize for the editorial errors. We have modified it. Please see the attachment Line 30.

Point 2. According to the authors and cited references the theme can provide innovative information.

Response: Thank you for your careful comment and compliments.

Point 3. The methodology is well explained.

Response: Thank you for your careful review and encouragement.

Point 4. The conclusions are in accordance with the proposed objectives and with the findings of the project.

Response: Thank you very much for your rigorous and meticulous review and acknowledgment of our manuscript.

Point 5. The figures presented in the text are well presented and provide information on important points of the project. Some corrections are necessary:Figure 6 -- correct:Shear rate Viscosity (Pa.s)

Response: Thank you for your critical comments. We have modified it. Please see the attachment Figure 6.

Reviewer 2 Report

Comments and Suggestions for Authors

I am sorry but the paper could not be accepted for publication in its current shape...First of all, the manuscript lacks clarity...quite a few sentences are long, complex, incomprehensible for the reader...The manuscriprt needs an extensive reformulation....besides:

- the innovation aspect of the paper should be further substantiated...authors should better explain (at the end of introduction section) what is its added value in this scientific field.

- Figures/tables missing...

- the right MDPI template should be used in the updated version (+right style for the list of References) 

- Can the authors explain (either in Material and methods or results sections) why the have choses these specific testing range (i) for XG proportions (ii) for NaCl concentrations?

Comments on the Quality of English Language

Extensive reformulation/editorial check is needed...

Author Response

Dear Reviewer,

      Thank you very much for your critical and valuable comments of reviewers concerning our manuscript entitled “Effect of ionic strength on the formation and stability of the ovalbumin-xanthan gum complex emulsions”(ID: foods-2753995. We have carefully revised the manuscript. All changes to the text have been made in the revised manuscript marked with red, so that you may easily identify them. We also responded point by point to you and reviewer’s comments as listed below. We hope the revised paper would satisfy you and the reviewer. We would appreciate you very much if you let us know your decision at your earliest convenience.

      Thank you very much again for your kind help.

Best wishes,

Na Wu

对审稿人的回应:

Point 1. First of all, the manuscript lacks clarity...quite a few sentences are long, complex, incomprehensible for the reader...

Response: Thank you for your careful review and we apologize for the lack of clarity in our presentation. After reading the entire text, we had modified the unclear, incomprehensible, and complex sentences as appropriate. 

  1. In recent years, proteins have been widely used as emulsifiers because of their excellent biocompatibility, unique charge properties, large molecular structure, and amphiphilicity. They can be adsorbed to the surface of droplets during the homogenization process, and stabilizing emulsions by generating electrostatic repulsion and strong spatial site resistance between particles. Please see the attachment Lines 49-53.
  2. For instance, the complexes of EWP-high methoxylated pectin, which ae synthesized via Maillard reactions, exhibit favorable emulsification properties and demonstrate stability for a duration of 40 weeks at ambient temperatur. This can be attributed to the heightened viscosity of the aqueous phase induced by pectin, as well as the exposure of hydrophobic protein groups through thermal treatment. Consequently, spatial repulsion and electrostatic interactions are facilitated, thereby enhancing the stability of the complex structure. Please see the attachment Lines 74-80.
  3. The cereal protein-XG emulsions prepared under the same conditions remain stable at high salt concentrations, while cereal protein-pectin emulsions are stable only at low salt concentrations.Please see the attachment Lines 97-99.
  4. For example, pH affects the degree of dissociation of the charged groups in the complex, thus changes their positive or negative charges and numbers, which in turn affects the stability of the emulsion. Please see the attachment Lines 105-107.
  5. Meanwhile, ionic strength played an important role in the degree of electrostatic binding and structure of protein-polysaccharide complexes. It’s mainly due to that salt ions can shade the charges of protein molecules, reduce the intermolecular repulsive forces and promote the cross-linking of proteins to form aggregates of larger particles. Thus, it can induce and accelerate the phase separation of protein-polysaccharide systems.Please see the attachment Lines 111-117.
  6. In order to clarify the influence of ionic strength on the interaction of the OVA-XG solution system, the protein conformation, UV absorption spectra were analyzed in this paper. However, since the UV absorption peaks of these three amino acids at 280 nm may overlap each other and be difficult to distinguish, a second-order derivative analysis of the UV absorption peaks was conducted. As shown in Fig. 2, the UV absorption peaks of the OVA-XG composite solution shifted to varying degrees at different ionic strengths as the concentration of XG increased. This suggests that the interaction between OVA and XG influenced the protein molecule's microenvironment, causing the displacement of amino acid residues. Please see the attachment Lines 383-291.
  7. When the maximum absorption wavelength (λ max) of the fluorescence emission peak shifts to a longer wavelength, it indicates that the amino acid residues are red shifted, meaning that more amino acid residues are exposed to the hydrophilic environment. Conversely, a blue shift occurs when tryptophan moves to a more hydrophobic environment. Please see the attachment Lines 304-308
  8. However, within a specific range of ionic strength, these interactions cause the emulsion to flocculate rapidly and form a network structure to stabilize the emulsion. At the same time, moderate flocculation is beneficial for the stability of the emulsion system.Please see the attachment Lines 348-352.
  9. It has been reported in the literature that emulsions exhibit good stability when the absolute zeta potential values exceed 30 mV. This is attributed to increased electrostatic repulsion between emulsion droplets, effectively preventing coalescence formation. Please see the attachment Lines392-395.
  10. This may be attributed to the negative charge of XG and OVA at pH 5.5, which increased the electrostatic repulsion between droplets and weakened the electrostatic mutual attraction effect. Meanwhile, the electrostatic shielding effect of NaCl would further strengthen this force and prevent droplet aggregation.Please see the attachment Lines 397-401.
  11. Secondly, XG is an anionic polysaccharide with a potential value of approximately -45mV in a saline solution. Therefore, excessive or high concentrations of XG may result in a reduction of the system's potential. However, it was observed that the OVA-0.3%XG emulsion exhibited the lowest potential (all lower than -55mV) at each ion concentration, possibly because of the presence of free XG alongside XG bound to protein in the system. Please see the attachment Lines413-419.
  12. As shown in Fig. 5, the dynamic interfacial tension of all samples showed a trend of rapid decrease and then equilibrium with increasing time. This suggests that the protein emulsions reduced the interfacial tension in a consistent manner, even at varying NaCl concentrations. Please see the attachment Lines 429-432.
  13. Furthermore, the inclusion of NaCl, comprising sodium and chloride ions, reduced the electrostatic repulsion between charged emulsions at the oil-water interface. This improved the stability of the protein adsorption layer and allowed NaCl and low concentrations of XG (0.1%wt, 0.2%wt) to work together to decrease the interfacial tension of the composite solutions synergistically. However, the interfacial tension of the OVA-0.3%XG emulsion increased with higher NaCl concentration, likely because of the elevated XG concentration and the formation of insoluble polymers by OVA-0.3%XG. The excessive XG hindered the adsorption of proteins at the oil-water interface, leading to an increase in interfacial tension.Please see the attachment Lines 444-452. 
  14. In the range of 0.1~100 s−1shear rate, the apparent viscosity of all samples decreased as the shear rate increased. The composite emulsions exhibited the shear thinning behavior characteristic of pseudoplastic fluids, indicating non-Newtonian fluid properties. Please see the attachment Lines 465-468.
  15. Interestingly, at low XG concentrations, the viscosity of OVA-XG emulsions formed with small amounts of salt ions (0.02M, 0.04M) showed a decreasing trend, while the viscosity of emulsions at high ion concentrations (0.08M) increased. This indicates that the composite emulsion formed by OVA-0.1% XG had a higher degree of inter-droplet flocculation at high salt concentrations compared to a low salt concentration.Please see the attachment Lines 477-482. 
  16. This indicates that the apparent viscosity of the OVA-XG complex emulsion is dependent on the concentration of XG. A sufficiently high amount of XG can inhibit OVA settling and aggregation, which is crucial for the practical application of the OVA-XG complex system, such as in fat substitution.Please see the attachment Lines 499-502.
  17. It is noteworthy that when various salt concentrations were added to the OVA-0.3% XG emulsion, the CI values were consistently 0 for each salt ion concentration. This indicates that the CI values of the composite emulsions were less influenced by salt ions at higher XG concentrations.Please see the attachment Lines 517-520.
  18. During storage, the molecular movement within the emulsion system accelerates, leading to the rearrangement or detachment of the protein molecules adsorbed at the oil-water interface, thereby altering the interfacial structure. This leads to a broader particle size distribution and increases the probability of coalescence.Please see the attachment Lines 525-529.
  19. As shown in Fig. 8, the microstructure of the single OVA emulsion revealed large fat globules with unevenly distributed gaps between lipid droplets, preventing the formation of a dense network structure and resulting in poor emulsion stability. As the concentration of salt ions increases, the large oil droplets gradually become smaller. This is likely due to the synergistic effect of OVA and NaCl, which reduces the interfacial tension and forms an interfacial protein film around the oil droplets. Consequently, this reduces the collision between molecules at the interface. Please see the attachment Lines 544-551.

Point 2. the innovation aspect of the paper should be further substantiated...authors should better explain (at the end of introduction section) what is its added value in this scientific field.

Response: Thank you for your rigorous and professional comments, according to your suggestion, we have added the innovation aspect and value in this scientific field as follows: We established the egg white protein-polysaccharide gel and emulsion system in the early stage, compared the effects of neutral polysaccharides (dextran and inulin) and anionic polysaccharides (xanthan gum) on the properties of egg white protein, and finally screened out xanthan gum, which proved that EWP-XG gels [1] and OVA-XG complex emulsions [2] can be used as protein-based fat replacers in foods such as low-fat ice cream and mayonnaise. However, the properties of emulsions are influenced by various factors in food, such as pH, ionic strength, and temperature, which can lead to instability in their application, thereby limiting their practical applications [3-5]. Therefore, it is essential to study the impact of external factors. Because egg white proteins are a complex system, ovalbumin, the most abundant protein in egg whites, was chosen for mechanistic studies. Previously, we have demonstrated the impact of pH and xanthan gum concentration on the stability of the emulsions. Our findings revealed that the particle size of the emulsions decreased gradually with increasing xanthan gum concentration, and the storage stability of the emulsions significantly improved. Meanwhile, the stability of the emulsion largely depends on the pH value, and the OVA-XG composite emulsion exhibits better stability at pH 5.5 [2]. However, salt plays an important role in food processing such as flavouring, preserving and moisturising. Meanwhile, the effect of NaCl on the stabilisation of emulsions by OVA-XG complexes has not been determined. Therefore, the present study focused on investigating the impact of ionic strength on the formation and stabilization of OVA-XG complex emulsions. The study also aimed to explore the potential mechanism by which ionic strength affects the emulsion stabilization of this complex. The ultimate goal was to establish a theoretical foundation for creating fat substitutes for stabilized OVA-XG emulsions, developing nutrient delivery carriers, and advancing the overall utilization of poultry eggs. 

Please see  the attachment Lines131-151.

References:

  1. Zhang H, Yang L, Tu Y, et al.Changes in texture and molecular forces of heated‐induced egg white gel with adding xanthan gum[J]. Journal of food process engineering. 2019, 42(4): e13071. https://doi.org/10.1111/jfpe.13071.
  2. Xiao, N., He, W., Zhao, Y., Yao, Y., Xu, M., Du, H., Wu, N., Tu, Y. Effect of pH and xanthan gum on emulsifying property of ovalbumin stabilized oil-in water emulsions. 2021, 147. https://doi.org/10.1016/j.lwt.2021.111621.
  3. He, W., Xiao, N., Zhao, Y., Yao, Y., Xu, M., Du, H., Wu, N., Tu, Y. Effect of polysaccharides on the functional properties of egg white protein: A review. Journal of Food Science. 2021, 86(3), 656-666. https://doi.org/10.1111/1750-3841.15651.
  4. Jia, W., Cui, B., Ye, T., Lin, L., Zheng, H., Yan, X., Li, Y., Wang, L., Liu, S., Li, B. Phase behavior of ovalbumin and carboxymethylcellulose composite system. Carbohydrate polymers. 2014, 109, 64-70. https://doi.org/10.1016/j.carbpol.2014.03.026
  5. Niu, F., Dong, Y., Shen, F., Wang, J., Liu, Y., Su, Y., Xu, R., Wang, J., Yang, Y. Phase separation behavior and structural analysis of ovalbumin–gum arabic complex coacervation. Food Hydrocolloids 2015, 43, 1-7. https://doi.org/10.1016/j.foodhyd.2014.02.009.

Point 3. Figures/tables missing...

Response: Thank you for your careful comment. All the data in the manuscript is presented as visual graphs rather than tables presenting the data. Please see the attachment Fig 1 to Fig8.

Point 4. the right MDPI template should be used in the updated version (+right style for the list of References) 

Response: Thank you for your careful review. We have re-read the MDPI submission guidelines carefully and double-checked the formatting of all references and corrected any errors. Please see the attachment Lines 608-728.

Point 5. Can the authors explain (either in Material and methods or results sections) why the have choses these specific testing range (i) for XG proportions (ii) for NaCl concentrations?

Response: Thank you very much for your rigorous and meticulous review. Previously, we compared the effects of neutral polysaccharides (dextran and inulin) and anionic polysaccharides (xanthan gum) on the properties of egg white proteins and found that sugar dextran and inulin weakened the gelation and rheology of egg white proteins. Low concentrations of xanthan gum reduced the intramolecular repulsion of egg white gels and significantly improved the texture of egg white protein gels. Therefore, XG was screened for further studies. Since egg white protein is a mixture of various proteins, in order to investigate the mechanistic interaction between EWP and XG, we chose OVA, which is the protein with the highest content in egg white. In the study of the effect of pH and xanthan gum concentration on OVA-XG composite solution, it was found that there was an electrostatic interaction between OVA and XG.The stability of OVA-XG composite emulsions was better at a pH of 5.5. With the increase of XG concentration (0%-0.4% wt), the particle size of the emulsion was gradually reduced, and the storage stability of the emulsion was significantly improved. However, when the XG concentration was increased to 0.04%, there was no significant difference from 0.03%. Secondly, salt ions shield the surface charges of proteins and polysaccharides, reducing the electrostatic interactions between them. Additionally, salt ions also shield the proteins themselves from electrostatic repulsion. 

同时,在卵清蛋白和多糖(如黄原胶、羧甲基纤维素和阿拉伯树胶)之间复合物形成的研究中,较低的氯化钠浓度(小于0.02 M)可用于通过增强它们的相互作用来促进复合物的形成。然而,较高的 NaCl 浓度(大于 0.02 M)会由于屏蔽效应而抑制复合物的形成。因此,选择NaCl(0-0.08 M)来研究所制备的OVA-XG复合乳液的稳定性。

要点 6. 对英语语言质量的评论

回复:非常感谢您提出的宝贵建议。我们现在已经在语言和可读性方面开展了工作,并且还让以英语为母语的人参与了语言纠正。我们真心希望流程和语言水平能够得到实质性的提高。

Reviewer 3 Report

Comments and Suggestions for Authors

1. Abstract " to show a significant stability." please consider for the meaning as it should be. There is no data indicating comparison numerical result with the use of single xanthan and OVA group in Abstract. 

2. Normally, there are many grade and type of xanthan gum with different properties so how the authors conclude for its specific effect on emulsion and with your concern.

3. First paragraph of Introduction please more review of previous reports on the effect of ionic strength influencing emulsion stabilizer properties and function with their explanation on mechanism or rationale.  Additionally, please include why ionic strength is come from during food processing?

4.  For Introduction for xanthan gum why don't you review on effect of ionic strength on physicochemical properties of this bacterium polymer of previous study both in field of foods or pharmaceuticals.?

5. Xanthomonas campestris should be italic and informed for its detail including M.W., % subsitution of each main functional groups and why do the authors selected this type.

6. The control groups using single stabilizer should be include in experiment.

7. Please provide the reason for using Fluorescence spectra measurements in this work. How is it important for explain anything crucially. If the data is not significant different they should removed.

8. What is the "s" in x-axis of fig 5,6? sec might be better.  There was no numerical data as Table in context.  pa.s ??? Why pH 5.5 is selected please include the reason in Introduction with supporting refs.

9. The presenting graph should include the S.D. as y error bat.

10. From Fig. 8, why is there no stabilizer appear on interface or surrounding oil droplets?

11. Please reveal the information about the instrument of The hanging drop method and have to measure the density before?

12.  More discussion with supporting references are needed for 3.2-3.6 by comparison with the related articles and theoretical aspect explanation especially role of xanthan gum concentration and ionic strength.

13. Discussion on type of ion on the physicochemical properties and change should be addressed.

14. The sample size of test should be added. 

Comments on the Quality of English Language

Fine with minor correction

Author Response

Dear Reviewer,

Thank you very much for your critical and valuable comments of reviewers concerning our manuscript entitled “Effect of ionic strength on the formation and stability of the ovalbumin-xanthan gum complex emulsions”(ID: foods-2753995). We have carefully revised the manuscript. All changes to the text have been made in the revised manuscript marked with red, so that you may easily identify them. We also responded point by point to you and reviewer’s comments as listed below. We hope the revised paper would satisfy you and the reviewer. We would appreciate you very much if you let us know your decision at your earliest convenience.

Thank you very much again for your kind help.

Best wishes,

Na Wu

Response to reviewers:

Point 1. Abstract " to show a significant stability." please consider for the meaning as it should be. There is no data indicating comparison numerical result with the use of single xanthan and OVA group in Abstract.

Response: Thank you for your careful review and suggestions, we have revised the meaning of the sentence as follows: The 0.08 M NaCl was added to the OVA-0.2% XG emulsion, which had a minimum particle size of 18.3 μm.

Please see the attachment line 22-23.

Point 2. Normally, there are many grade and type of xanthan gum with different properties so how the authors conclude for its specific effect on emulsion and with your concern.

Response: Thank you for your careful review. Firstly, xanthan gum is primarily obtained from Xanthomonas spp [1-3]. And the average molecular mass of XG depends on the specific strain and biofermentation conditions, while their chemical composition is mainly influenced by the choice of strain and medium composition [4]. Secondly, Xanthan gum solution has good salt resistance and acid and alkali resistance, etc. The influence of salt ions on xanthan gum is primarily associated with the concentration of xanthan gum, ionic species, concentration, and ionic valence. The cation primarily interacts with the glucuronide group on the side chain of xanthan gum, maintaining the viscosity of xanthan gum stable [1]. Xanthan gum has been selected for creating stable emulsions with various proteins because of its stability, emulsifying properties, thickening ability, suspension characteristics, and safety. Xanthan gum, derived from Xanthomonas spp, is commonly used as a thickener, emulsifier, and stabilizer in food research [5-7].

 We conducted a comparison of the impacts of two neutral polysaccharides (dextran and inulin) and an anionic polysaccharide (xanthan gum) on egg white gels [8]. Our findings revealed that the neutral polysaccharides weakened the gelation and rheological properties of egg white proteins, while a low concentration of xanthan gum notably enhanced the texture of egg white protein gels. Subsequently, we constructed OVA-XG composite emulsions and observed electrostatic interaction forces between them. In addition, the particle size of the emulsions decreased as the XG concentration increased, and the storage stability of the emulsions was significantly improved [9]. Our results suggest that XG has the potential to be used as an emulsion stabilizer for egg-white derived foods.

References:

  1. Habibi, H., Khosravi-Darani, K. Effective variables on production and structure of xanthan gum and its food applications: A review. Biocatalysis and Agricultural Biotechnology. 2017,10,130-140. https://doi.org/10.1016/j.bcab.2017.02.013.
  2. Krishna Leela J, Sharma G. Studies on xanthan production from Xanthomonas campestris[J]. Bioprocess Engineering. 2000,23:687-689.  https://doi.org/10.1007/s004499900054.
  3. Davidson I W. Production of polysaccharide by Xanthomonas campestris in continuous culture[J]. FEMS Microbiology Letters,1978:347-349. https://doi.org/10.1016/0378-1097(78)90024-1.
  4. Esgalhado M E, Roseiro J C, Collaço M T A. Interactive effects of pH and temperature on cell growth and polymer production by Xanthomonas campestris[J]. Process Biochemistry,1995:667-671. https://doi.org/10.1016/0032-9592(94)0044-1.
  5. Zapata Noreña C P, Bayarri S, Costell E. Effects of xanthan gum additions on the viscoelasticity, structure and storage stability characteristics of prebiotic custard desserts[J]. Food biophysics,2015,10:116-128.https://doi.org/10.1007/s11483-014-9371-2.
  6. Chaiya B, Pongsawatmanit R, Prinyawiwatkul W. Optimisation of wheat flour‐based sponge cake formulation containing tapioca starch and xanthan gum[J]. International Journal of Food Science & Technology,2015,50:532-540.https://doi.org/10.1111/ijfs.12706.
  7. Kumar A, Rao K M, Han S S. Application of xanthan gum as polysaccharide in tissue engineering: A review[J]. Carbohydrate Polymers. 2018,180:128-144. https://doi.org/10.1016/j.carpol.2017.10.009.
  8. Zhang H, Yang L, Tu Y, et al. Changes in texture and molecular forces of heated‐induced egg white gel with adding xanthan gum[J]. Journal of food process engineering. 2019(4): e13071. https://doi.org/10.1111/jfpe.13071.
  9. Xiao, N., He, W., Zhao, Y., Yao, Y., Xu, M., Du, H., Wu, N., Tu, Y. Effect of pH and xanthan gum on emulsifying property of ovalbumin stabilized oil-in water emulsions. 2021,147. https://doi.org/10.1016/j.lwt.2021.11621.

Point 3. First paragraph of Introduction please more review of previous reports on the effect of ionic strength influencing emulsion stabilizer properties and function with their explanation on mechanism or rationale Additionally, please include why ionic strength is come from during food processing?

Response: Thank you for your critical comments. We further examine the previous reports on the impact of ionic strength on the performance and function of emulsion stabilizers in the third paragraph of the preface and elucidate the underlying mechanism or principle as follows:

Because ionic strength (NaCl) is a crucial factor in flavouring, preservation, and quality of protein-based food and beverage products. It alters protein-protein, protein-polysaccharide, and protein-water interactions during processing, making it a significant factor in food processing.

 Ionic strength played an important role in the degree of electrostatic binding and structure of protein-polysaccharide complexes. Because ionic strength can induce and accelerate the phase separation of protein-polysaccharide systems [1]. It’s mainly due to that salt ions can shade the charges of protein molecules, reduce the intermolecular repulsive forces and promote the cross-linking of proteins to form aggregates of larger particles. Thus, it can induce and accelerate the phase separation of protein-polysaccharide systems [1]. For example, the addition of salt ions to egg white protein/carrageenan and lysozyme/carrageenan mixtures inhibits the electrostatic interactions between the protein and the carrageenan, which in turn shifts the critical pH transition point to a lower pH value [2]. Furthermore, salt ions can compete with proteins or polysaccharides with charged groups, affecting protein-polysaccharide interaction forces. Niu Fuge et al. found that when NaCl was added to the mixture, the anion competes with the negative charge binding site of the Arabic gum chain to bind to OVA [3]. Likewise, the cation competes with the positive charge binding site of OVA for binding to Arabic gum, thus reducing the interaction between OVA and Arabic gum [3]. That is, in the presence of salt ions, there is a reduction in complex formation. Different concentrations of ionic strength have an effect on the electrostatic interactions between the egg white - polysaccharide, thus interfering with complex formation [4]. Therefore, it was necessary to investigate the effect of ionic strength on emulsion stability. Please see the attachment Lines111-120.

References:

  1. Jia, W., Cui, B., Ye, T., Lin, L., Zheng, H., Yan, X., Li, Y., Wang, L., Liu, S., Li, B. Phase behavior of ovalbumin and carboxymethylcellulose composite system. Carbohydrate polymers. 2014,109,64-70. https://doi.org/10.1016/j.carbpol.2014.03.026.
  2. He, W., Xiao, N., Zhao, Y., Yao, Y., Xu, M., Du, H., Wu, N., Tu, Y. Effect of polysaccharides on the functional properties of egg white protein: A review. Journal of Food Science. 2021,656-666. https://doi.org/10.1111/1750-3841.15651.
  3. Niu, F., Dong, Y., Shen, F., Wang, J., Liu, Y., Su, Y., Xu, R., Wang, J., Yang, Y. Phase separation behavior and structural analysis of ovalbumin–gum arabic complex coacervation. Food Hydrocolloids 2015,43,1-7. https://doi.org/10.1016/j.foofhyd.2014.02.009.
  4. Fu, D., Deng, S., McClements, D. J., Zhou, L., Zou, L., Yi, J., Liu, C., Liu, W. Encapsulation of β-carotene in wheat gluten nanoparticle-xanthan gum-stabilized Pickering emulsions: Enhancement of carotenoid stability and bioaccessibility. Food Hydrocolloids. 2019,89, 80-89. https://doi.org/10.1016/j.foodhyd.2018.10.032.

Point 4. For Introduction for xanthan gum why don't you review on effect of ionic strength on physicochemical properties of this bacterium polymer of previous study both in field of foods or pharmaceuticals.?

Response: Thank you for your professional suggestion, we have added some information about these aspects as follows: Xanthan gum solution can enhance the viscosity of the system, thereby improving the emulsification of ovalbumin. This is attributed to its low concentration and high viscosity characteristics (the viscosity of a 1% aqueous solution is equivalent to 100 times that of gelatin) [1,2]. In addition, aqueous xanthan gum solutions exhibit good stability in extreme environments, such as varying pH levels and high salt concentrations [2]. The effect of salt on food applications of xanthan gum is primarily associated with the concentration of gum, ionic species, concentration, and ionic valence in foods. For instance, monovalent salts (NaCl), divalent metal salts( MgCl2、CaCl2) cause a slight decrease in viscosity [5]. Meanwhile, hydrated xanthan gum exhibits good salt resistance, allowing for the addition of up to 20-30% of salts without adversely affecting viscosity [5]. Moreover, Xanthan gum is commonly used in the food, medicine, and chemical industries due to its stability, emulsification, thickening, suspension, and safety.

Please see the attachment Lines 80-93.

References:

  1. Wang C S, Natale G, Virgilio N, et al. Synergistic gelation of gelatin B with xanthan gum[J]. Food Hydrocolloids,2016,60:374-383. https://doi.org/10.1016/j.foodhyd.2016.03.043.
  2. Patel J, Maji B, Moorthy N S H N, et al. Xanthan gum derivatives: Review of synthesis, properties and diverse applications[J]. RSC advances,2020:27103-27136. https://pubs.rsc.org/en/content/articlepdf/2020/ra/d0ra04366d.
  3. Higiro J, Herald T J, Alavi S, et al. Rheological study of xanthan and locust bean gum interaction in dilute solution: Effect of salt[J]. Food Research International,2007,40:435-447.https://doi.org/10.1016/j.foodres.2006.02.002
  4. Habibi, H., Khosravi-Darani, K. Effective variables on production and structure of xanthan gum and its food applications: A review. Biocatalysis and Agricultural Biotechnology. 2017,10,130-140. https://doi.org/10.1016/j.bcab.2017.02.013

Point 5. Xanthomonas campestris should be italic and informed for its detail including M.W., % subsitution of each main functional groups and why do the authors selected this type.

Response:Thank you for your rigorous and professional comments, according to your suggestion, we have modified it and added some information about these aspects. Our rationale for selecting xanthan gum (XG) derived from Xanthomonas campestris is as follows: Firstly, after reviewing the relevant literature, we discovered that xanthan gum is primarily obtained from Xanthomonas spp. Such as Xanthomonas campestris, Xanthomonas pelargonii, Xanthomonas phaseoli and Xanthomonas malvacearum during aerobic fermentation [1,2]. The average molecular mass of XG depends on the specific strain and biofermentation conditions, while their chemical composition is mainly influenced by the choice of strain and medium composition. Secondly, among the microbial gum, XG is highly pseudoplastic and viscous [3,4]. It is compatible with most salts and resistant to acids, alkalis, and high temperatures, among other properties. Secondly, our initial studies showed the potential of this XG type as an emulsion stabilizer for foods derived from egg whites [5].

Please see the attachment Lines 80-85,159.

References:

  1. Habibi, H., Khosravi-Darani, K. Effective variables on production and structure of xanthan gum and its food applications: A review. Biocatalysis and Agricultural Biotechnology. 2017,10,130-140. https://doi.org/10.1016/j.bcab.2017.02.013. 
  2. Krishna Leela J, Sharma G. Studies on xanthan production from Xanthomonas campestris[J]. Bioprocess Engineering. 2000,23:687-689.  https://doi.org/10.1007/s004499900054.
  3. Li, P., Zeng, Y., Xie, Y., Li, X., Kang, Y., Wang, Y., Zhang, Y. Effect of pretreatment on the enzymatic hydrolysis of kitchen waste for xanthan production[J]. Bioresource Technology. 2017,223:84-90. https://doi.org/10.1026/j.biortech.2016.10.035. 
  4. Higiro J, Herald T J, Alavi S, et al. Rheological study of xanthan and locust bean gum interaction in dilute solution: Effect of salt[J]. Food Research International,2007:435-447. https://doi.org/10.1016/j.foodres.2006.02.002.
  5. Xiao, N., He, W., Zhao, Y., Yao, Y., Xu, M., Du, H., Wu, N., Tu, Y. Effect of pH and xanthan gum on emulsifying property of ovalbumin stabilized oil-in water emulsions. Lwt 2021,147. https://doi.org/10.1016/j.lwt.2021.111621.

Point 6. The control groups using single stabilizer should be include in experiment.

Response:Thank you for your considerate review.We prepared OVA-XG composite emulsions with varying concentrations of xanthan gum, using ovalbumin as a emulsifier, and investigated the impact of different ionic concentrations (0-0.08 M NaCl) on the emulsions. Where the concentration of xanthan is 0, it serves as the experimental group as a sole stabilizer. Therefore, no other stabilisers were added in our experiments.

Point 7. Please provide the reason for using Fluorescence spectra measurements in this work. How is it important for explain anything crucially. If the data is not significant different they should removed.

Response:Thank you for your careful review.This is because fluorescence spectroscopy measurements can be used to determine whether the interaction of OVA with XG at different ion concentrations results in structural changes in OVA. From our fluorescence spectroscopy results, we observed an increase in fluorescence intensity from 853 nm to 911 nm as the NaCl concentration increased in the absence of 0.0% XG-OVA. Meanwhile, the λ max of the composite solution systems of 0.1% XG-OVA,0.2%XG-OVA, and 0.3% XG-OVA varied from 343 nm to 331 nm,341nm to 336 nm, and 339 nm to 337 nm, respectively. A low salt concentration enhances the electrostatic interaction between OVA and XG, while a high salt concentration hinders the formation of the composite. Meanwhile, the addition of XG and NaCl altered the structure of OVA to different extents, leading to variations in the microenvironment of amino acids.

Point 8. What is the "s" in x-axis of fig 5,6? sec might be better.  There was no numerical data as Table in context.  pa.s ??? Why pH 5.5 is selected please include the reason in Introduction with supporting refs.

Response: Thank you for your careful comment. The "s" on the x-axis in Figs. 5 and 6 represent units of time [1] and shear rate, respectively [2]. Because based on our previous studies, the results confirmed emulsions stability largely depended on the pH value. OVA-XG composite emulsion showed a better stability at pH 5.5. Please see the attachment Lines142-146.

References:

  1. Cai Y, Deng X, Liu T, et al. Effect of xanthan gum on walnut protein/xanthan gum mixtures, interfacial adsorption, and emulsion properties[J]. Food Hydrocolloids,2018,79:391-398. https://doi.org/10.1016/j.foodhyd.2018.01.006.
  2. Guo B, Hu X, Wu J, et al. Soluble starch/whey protein isolate complex-stabilized high internal phase emulsion: Interaction and stability[J]. Food Hydrocolloids,2021,111:106377. https://doi.org/10.1016/j.foodhyd.2020.106377.

Point 9. The presenting graph should include the S.D. as y error bat.

Response: Thank you for your considerate review. Fig. 1, 4 and 7 are made with corresponding error bars. And Fig. 2, Fig. 3, Fig. 5, Fig. 6, Fig. 7 due to the type of the graphs are Y-offset stacked line graphs. All measurements were performed at least three times in parallel, in line with statistical analyses, and we can provide the raw data as a reference.

Point 10. From Fig. 8, why is there no stabilizer appear on interface or surrounding oil droplets?

Response: Thank you for your careful review. Since we utilize ovalbumin as an emulsifier to create stable OVA-XG oil-in-water emulsions, no additional emulsifiers are included. As a result, no oil droplets or stabilizers are found around the interface in Fig. 8. However, as shown in Fig. 8, we successfully constructed the OVA-XG oil-in-water emulsion. OVA and NaCl synergistically reduce the interfacial tension and stabilize the oil droplets by minimizing the collision between them. Meanwhile, XGcan stabilize the emulsion by increasing the viscosity within the system, which slows down the intermolecular forces.

Point 11. Please reveal the information about the instrument of The hanging drop method and have to measure the density before?

Response: Thank you for your positive suggestions. We apologize for the trouble caused by our negligence, and we have revised the information about the instrument of the hanging drop method based on your suggestion. Please see the attachment Lines 202-204.

Point 12. More discussion with supporting references are needed for 3.2-3.6 by comparison with the related articles and theoretical aspect explanation especially role of xanthan gum concentration and ionic strength.

Response: Thank you for your rigorous and professional comments, according to your suggestion,we have provided more discussion and supporting literature for 3.2-3.6. 

Revise section 3.2 as follows:

As shown in Fig. 2, the UV absorption peaks of the OVA-XG composite solution shifted to varying degrees at different ionic strengths as the concentration of XG increased. This suggests that the interaction between OVA and XG influenced the protein molecule's microenvironment, causing the displacement of amino acid residues. Please see the attachment Lines 287-291.

From Fig. 2, it can be observed that the addition of a small amount of salt ions (0.02 M) leads to an increase in the absorbance of the solution. It is possible that the protein molecule unfolds, leading to an increase in the chromogenic groups on the surface and subsequently an increase in absorbance. Please see the attachment Lines 299-302.

Secondly, after adding polysaccharides and salt, the protein structure was disrupted to some extent. This caused the hydrophobic amino acids to be exposed and gathered on the protein surface, leading to the exposure of the chromogenic group to the solvent and a subsequent reduction in fluorescence intensity [1]. Please see the attachment Lines 331-335.

Revise section 3.3 as follows:

However, within a specific range of ionic strength, these interactions cause the emulsion to rapidly flocculate and form a network structure to stabilize the emulsion [2]. At the same time, moderate flocculation is beneficial for the stability of the emulsion system [3]. Please see the attachment Lines 348-352.

Moreover, the addition of salt ions alters the osmotic pressure change of the system, which in turn affects the particle size [4]. For example, osmotic pressure can reduce the swelling capacity of OVA-XG complexes in salt solutions, resulting in increased aggregation and precipitation [4]. Please see the attachment Lines 362-365.

Revise section 3.4 as follows:

This could be attributed to the gradual increase of XG led to the distribution of a large number of polysaccharides with negatively charged groups in the emulsion. This increased the electrostatic repulsion of the molecules and effectively prevented the proximity and aggregation of the emulsion particles, thereby enhancing the stability of the emulsion [5]. Please see the attachment Lines 406-410.

Revise section 3.5 as follows:

Furthermore, the inclusion of NaCl, comprising sodium and chloride ions, reduced the electrostatic repulsion between charged emulsions at the oil-water interface. This improved the stability of the protein adsorption layer and allowed NaCl and low concentrations of XG (0.1%wt,0.2%wt) to work together to decrease the interfacial tension of the composite solutions synergistically [6]. However, the interfacial tension of the OVA-0.3%XG emulsion increased with higher NaCl concentration, likely because of the elevated XG concentration and the formation of insoluble polymers by OVA-0.3%XG. The excessive XG hindered the adsorption of proteins at the oil-water interface, leading to an increase in interfacial tension [7]. Please see the attachment Lines 443-452.

Revise section 3.6 as follows:

At a lower shear rate, the viscosity of the OVA emulsion increased after the addition of NaCl, indicating that the interaction between the emulsion particles was enhanced due to the addition of sodium ions (Na+) [8]. Please see the attachment Lines 471-474.

This indicates that the apparent viscosity of the OVA-XG complex emulsion is dependent on the concentration of XG. A sufficiently high amount of XG can inhibit OVA settling and aggregation, which is crucial for the practical application of the OVA-XG complex system, such as in fat substitution. Please see the attachment Lines 499-502.

References:

  1. Zhang, l., Lin, W.-F., Zhang, Y., Tang, C.-H. New insights into the NaCl impact on emulsifying properties of globular proteins. Food Hydrocolloids. 2022,124. https://doi.org/10.1016/j.foodhyd.2021.107342.
  2. Lai, H., Zhan, F., Wei, Y., Zongo, A. W. S., Jiang, S., Sui, H., Li, B., Li, J. Influence of particle size and ionic strength on the freeze-thaw stability of emulsions stabilized by whey protein isolate. Food Science and Human Wellness. 2022,11,922-932. https://doi.org/10.1016/j.fshw.2022.03.018.
  3. Zhu, Y., McClements, D. J., Zhou, W., Peng, S., Zhou, L., Zou, L., Liu, W. Influence of ionic strength and thermal pretreatment on the freeze-thaw stability of Pickering emulsion gels. Food Chemistry. 2020,303,125401. https://doi.org/10.1016/j.foodchem.2019.125401.
  4. Melanie, H., Taarji, N., Zhao, Y., Khalid, N., Neves, M. A., Kobayashi, I., Tuwo, A., Nakajima, M. Formulation and characterisation of O/W emulsions stabilised with modified seaweed polysaccharides. International Journal of Food Science & Technology. 2019,55,211-221. http://doi.org/10.111/ijfs.14264.
  5. Qiu, C., Zhao, M., McClements, D.J. Improving the stability of wheat protein-stabilized emulsions: Effect of pectin and xanthan gum addition. Food Hydrocolloids,2015,43:377-387. https://doi.org/10.1016/j.foodhyd.2014.06.013.
  6. Kontogiorgos, V. Polysaccharides at fluid interfaces of food systems. Advances in colloid and interface science. 2019,270,28-37. https://doi.org/10.1016/j.cis.2019.05.008.
  7. Sun, C., Gunasekaran, S., Richards, M. P. Effect of xanthan gum on physicochemical properties of whey protein isolate stabilized oil-in-water emulsions. Food Hydrocolloids 2007,555-564. https://doi.org/10.1016/j.foodhyd.2006.06.003.
  8. Tu, Y., Zhang, X., Wang, L. Effect of salt treatment on the stabilization of Pickering emulsions prepared with rice bran protein. Journal of Food Science and Technology. 2023, 166, 112537. https://doi.org/10.1016/j.foodres.2023.112537.

Point 13. Discussion on type of ion on the physicochemical properties and change should be addressed.

response: Thank you for your considerate review. We investigated the effect of different ionic strengths (0-0.08 M) on OVA-XG composite solution system and composite emulsion system. The results showed that the addition of different concentrations of salt ions increased or decreased their electrostatic or hydrophobic interactions, and promoted or inhibited protein refolding and aggregation, thereby affecting the OVA-XG complex solution. Secondly, the addition of XG can increase the viscosity and salt can improve its interfacial adsorption capacity, and XG and salt can synergistically improve the stability of the emulsion under specific conditions. In the presence of XG, the rheology of OVA-XG composite emulsions were characterized by shear thinning properties, and a small amount of salt ions weakens the interaction force between droplets. With the increase of salt concentration, the network structure formed inside the emulsion, the absolute value of zeta potential increased significantly, the particle size of the emulsion stabilized by OVA-XG complex gradually decreased, and the seven-day storage stability of the emulsion improved.

Point 14. The sample size of test should be added.

Response: Thank you for your positive suggestions. We apologize for the trouble caused by our negligence.We have added the sample size to the diagram. Please see the attachment Fig 8.

Point 15. Comments on the Quality of English Language

Response: Thank you very much for your valuable suggestion. We have now worked on both language and readability and have also involved native English speaker for language corrections.We really hope that the flow and language level have been substantially improved.

Round 2

Reviewer 2 Report

Comments and Suggestions for Authors

No more comments

Author Response

Dear Reviewer,

Thank you very much for your kind acknowledgement of our revised manuscript "Effect of ionic strength on the formation and stability of ovalbumin-xanthan gum composite emulsion" (ID: food-2753995). Thank you again for your professional review of our manuscript and for your constructive comments and valuable suggestions.

Thank you very much again for your kind help.

Best wishes,

Na Wu